# Vaginal ring acceptability and related preferences among women in low- and middle-income countries: A systematic review and narrative synthesis

Jennifer B. Griffin[1], Kathleen Ridgeway[2], Elizabeth Montgomery[1], Kristine Torjesen[2], Rachel Clark[3], Jill Peterson[2], Rachel Baggaley[4], Ariane van der Straten[1,5]*

**1** RTI International, Center for Global Health, Durham, NC, United States of America, **2** FHI 360, Global Health Population and Nutrition, Durham, NC, United States of America, **3** RTI International, Public Health Research Division, Durham, NC, United States of America, **4** World Health Organization, Geneva, Switzerland, **5** University of California, San Francisco, Department of Medicine, San Francisco, CA, United States of America

* ariane@rti.org

**Data Availability Statement:** All relevant data are within the paper and its Supporting Information files.

## Abstract

The vaginal ring (VR) is a female-initiated drug-delivery platform used for different indications, including HIV pre-exposure prophylaxis (PrEP). We conducted a systematic review of VR acceptability, values and preferences among women in low- and middle-income countries (LMIC) to inform further investment and/or guidance on VR use for HIV prevention. Following PRISMA guidelines, we used structured methods to search, screen, and extract data from randomized controlled trials (RCTs) and observational studies reporting quantitative outcomes of acceptability of the VR for any indication published 1/1970-2/2019 (PROSPERO: CRD42019122220). Of 1,110 records identified, 68 met inclusion criteria. Studies included women 15–50+ years from 25 LMIC for indications including HIV prevention, contraception, abnormal bleeding, and menopause. Overall VR acceptability was high (71–98% across RCTs; 62–100% across observational studies), with 80–100% continuation rates in RCTs and favorable ease of insertion (greater than 85%) and removal 89–99%). Users reported concerns about the VR getting lost in the body (8–43%), although actual expulsions and adverse events were generally infrequent. Most women disclosed use to partners, with some worrying about partner anger/violence. The VR was not felt during intercourse by 70–92% of users and 48–97% of partners. Acceptability improved over time both within studies (as women gained VR experience and worries diminished), and over chronological time (as the device was popularized). Women expressed preferences for accessible, long-acting, partner-approved methods that prevent both HIV and pregnancy, can be used without partner knowledge, and have no impact on sex and few side effects. This review was limited by a lack of standardization of acceptability measures and study heterogeneity. This systematic review suggests that most LMIC women users have a positive view of the VR that increases with familiarity of use; and, that many would consider the VR an acceptable future delivery device for HIV prevention or other indications.

**Funding:** Co-authors JG, EM, and AS were supported by TIP, a program made possible by the generous support of the American people through the U.S. President's Emergency Plan for AIDS Relief (https://www.pepfar.gov/) to RTI International under the terms of the Grant No. AID-OAA-A-14-00012. Co-authors KR, KT, and JP were supported by the OPTIONS Consortium, a program made possible by the generous assistance from the American people through the U.S. Agency for International Development (USAID) and the U.S. President's Emergency Plan for AIDS Relief (PEPFAR). Financial assistance was provided by USAID (https://www.usaid.gov/) to FHI 360, the Wits Reproductive Health and HIV Institute, and AVAC under the terms of Cooperative Agreement No. AID-OAA-A-15-00035. The contents do not necessarily reflect the views of USAID, PEPFAR, or the United States Government. The funders did not play any role in the study design, data collection and analysis, decision to publish, or preparation of the manuscript.

**Competing interests:** The authors have declared that no competing interests exist.

**Abbreviations:** API, Active pharmaceutical ingredient; COC, Combined oral contraceptive; LMIC, Low- and middle-income countries; MeSH, Medical subject headings; PEPFAR, U.S. President's Emergency Plan for AIDS Relief; PrEP, Pre-exposure prophylaxis; PRISMA, Preferred Reporting Items for Systematic Reviews and Meta-Analysis; RCT, Randomized controlled trial; USAID, U.S. Agency for International Development; MPT, Multi-purpose prevention technology; VR, Vaginal ring.

## Introduction

The vaginal ring (VR) is a long-acting drug-delivery platform that diffuses drugs embedded in polymeric matrices into the vaginal epithelium [1]. By avoiding gastrointestinal absorption and first-pass hepatic metabolism, VRs provide sustained, therapeutic levels of drugs with lower systemic exposure compared to oral therapies, possibly resulting in fewer side effects [1, 2].

VRs have demonstrated efficacy and are marketed for multiple indications including contraception (NuvaRing®) [3], progesterone-only contraception in nursing mothers (Progering®) [4], management of genitourinary syndrome and vasomotor symptoms of menopause (Estring®, Femring®, and Fertiring®) [5, 6], and polycystic ovarian syndrome [7] among others. More recently, a dapivirine ring for HIV prevention demonstrated moderate efficacy for HIV pre-exposure prophylaxis (PrEP) [8, 9] and is under review by the European Medicines Agency, with upcoming regulatory submissions to the US Food and Drug Administration and South African Health Products Regulatory Authority. Other VRs for HIV PrEP and multi-purpose prevention technologies (MPTs) are at various stages in the development pipeline [8, 10–13]. MPT VRs are particularly promising, as they increase efficiencies for users and health systems by simultaneously addressing multiple sexual and reproductive health needs [14].

Given the high burden of unintended pregnancies and sexually transmitted infections, including HIV, research efforts have focused on the development and implementation of female-initiated drug-delivery methods, such as the VR, particularly for low- and middle-income countries (LMIC). Improving options for female-initiated platforms in sexual and reproductive health is critical to address women's diverse and dynamic preferences [15]. The addition of one contraceptive method to half or more of a population has been demonstrated to increase overall contraceptive use by 4 to 8% [16]. The correlation between increased platform options and uptake is also hypothesized to apply to HIV PrEP [17].

Acceptability and preference research play a crucial role in the design, evaluation, and implementation of VRs and other sexual and reproductive health products, as well as in the development of associated clinical guidance and policies [18]. To our knowledge, there has not been a review on the acceptability and preferences for VRs as a drug-delivery platform, irrespective of active pharmaceutical ingredient (API). We conducted a systematic review of the evidence base in the published and grey literature to assess the acceptability of the VR and related preferences among women in LMIC to inform further investment and/or guidance on VRs for HIV prevention.

## Methods

This systematic review adheres to Preferred Reporting Items for Systematic Reviews and Meta-Analysis (PRISMA) guidelines [19] and the protocol is registered with PROSPERO (ID: CRD42019122220).

### Conceptual framework

There is no consensus in the literature regarding the definition and conceptualization of acceptability; however, recent attempts to develop theoretical frameworks for health-care intervention acceptability are facilitating more robust assessments of acceptability and its impacts on health outcomes [20, 21]. These frameworks recognize that acceptability is a multi-faceted concept, including attitudes, tolerances, and preferences, and is a key driver of intervention behavior. We operationalized the constructs from the Sekhon acceptability model to identify corresponding outcomes reported in the literature reviewed (Table 1) [21].

**Table 1. Sekhon acceptability model constructs and systematic review operationalization.**

| Construct | Operationalization | Corresponding outcomes |
|---|---|---|
| Affective attitude | Feelings about the intervention | Acceptability, liking or recommending; finding physical attributes acceptable |
| Burden | Perceived effort required to engage in intervention | Ease of use; ease of insertion and removal; cognitive and emotional burden of use |
| Ethicality | Intervention fit with an individual's value system; normative fit | Disclosure of use; use without partner/family knowledge; partner/family approval |
| Intervention coherence | Understanding of the intervention | We did not include studies reporting intervention coherence |
| Opportunity costs | Extent to which benefits, profits, or values must be given up engaging in the intervention | Impacts on sexual intercourse; vaginal discharge/irritation; expulsions; discomfort; and foreign body sensation |
| Perceived effectiveness | Extent to which the intervention is perceived to achieve its purpose | Perceived ability to prevent pregnancy/infectious disease/other VR outcomes; reduced risk of certain cancers |
| Self-efficacy | Confidence to perform the behaviors required for the intervention | Ability to support use |

### Eligibility criteria

We included RCTs and observational studies in the peer-reviewed and grey literature reporting quantitative VR acceptability and/or preference data, from World Bank classified LMIC published between January 1, 1970 and March 1, 2019, and available in English. We focused this review on quantitative studies due to the relatively complex and poorly defined methods around the integration of qualitative and quantitative data [22]. We excluded qualitative studies, secondary analyses, or studies that only reported effectiveness or API-related side effects or complications. Women of any age were included, as well as their partners if partner-reported acceptability data were reported. We included both hypothetical (i.e. discrete choice experiment with hypothetical products) studies as well as those that used active or placebo VRs with or without comparator products.

### Outcome measures

The primary outcome measure was VR acceptability, with acceptability defined as 'a multi-faceted construct that reflects the extent to which people delivering or receiving a healthcare intervention consider it to be appropriate, based on anticipated or experienced cognitive and emotional responses to the intervention' [21]. We considered 'accept,' 'satisfy,' 'like,' and 'would recommend,' as primary acceptability outcomes, corresponding to the 'affective attitude' construct (Table 1). We examined additional acceptability outcome measures corresponding to additional constructs, such as continuation, perceived effectiveness, non-API side effects, comfort, opinion on access, setting, frequency, route of administration, duration, and cost, impact on sex, partner acceptability, and stated preference (or ranking) or discrete choice compared to other options, as outlined in Table 1. In order to investigate the acceptability of the VR platform and not the API, we did not include known API side effects, with the exception of reports of bleeding if they reflected differences in route of administration or dose effects. Next, we extracted data regarding VR continuation, a frequently used proxy for acceptability in RCTs. Finally, we extracted data regarding women's values and preferences, which we defined as the relative importance that women place on benefits and burdens related to the device(s) under investigation. Values and preferences could include perceived effectiveness, route of administration, side effects, access/setting, frequency/duration, cost, comfort, impact on sex, and partner acceptability, among others. We also included fully contextualized or 'global' preference assessments of the VR in comparison to other delivery platforms which

included stated preference (or ranking) of one device over another or discrete choice experiments.

## Search strategy

We searched PubMed, Web of Science, and Embase from January 1, 1970 to March 1, 2019 using medical subject headings (MeSH) and keywords for relevant peer reviewed and grey literature. The complete search strategy can be found in supporting materials (S1 File). We hand-searched bibliographies of manuscripts and grey literature that met eligibility criteria to identify further, relevant references, which were subject to the same screening and selection process. We conducted a web search of VR acceptability to identify additional conference proceedings and reports. Finally, a predetermined list of VR research experts was consulted to identify additional grey literature not found through the above processes.

## Data screening and extraction

Two team members independently reviewed the reference list of articles for inclusion (JG and KR) using a three-stage approach, including title, abstract, and full text review. Discrepancies during the screening process were resolved via consensus. A primary reviewer extracted article information (KR, RC, or JP) and a secondary reviewer (KR or RC) checked for accuracy using structured data extraction tables in Microsoft Word to extract article information, descriptive data, methods and study design, and outcomes (S2 and S3 Files).

## Risk of bias and heterogeneity

We assessed risk of bias across randomized studies using Cochrane Collaboration methods by one primary and one secondary reviewer (KR and RC) against key criteria [23, 24]. The following judgments were used: low risk, high risk, unclear (i.e. lack of information or uncertainty over bias potential). Risk of bias across non-randomized studies were similarly assessed [25]. Conflicts were resolved via consensus. Studies with no 'high risk of bias' ratings were considered low risk of bias; studies with any 'high risk of bias' ratings were considered moderate risk of bias; and, two or more 'high risk of bias' ratings were considered high risk of bias. No studies were excluded based on risk of bias; however, we did compare acceptability outcomes in low risk versus and high/unclear risk studies.

We considered methodological and clinical heterogeneity by exploring the acceptability of the vaginal ring against both study design (RCT vs observational studies), and different subgroup participant characteristics (e.g. age, parity, etc. . .), as well as by the purpose of ring use. As data allowed, we presented comparative results on acceptability and preferences for studies that compared vaginal rings to other contraceptive, HIV/ID prevention, and menopausal management technologies.

## Data analysis

We tabulated results by study type and indication for VR use. Summary measures included means and percentages for descriptive acceptability outcomes, and odds ratios and risk ratios for comparative outcomes. We conducted a narrative synthesis of included studies, including a summary of the following: 1) VR acceptability outcomes; 2) VR continuation (a common proxy for acceptability); and, 3) related values and preferences. Analysis of acceptability was examined by VR indication (e.g. contraception, HIV prevention, etc.) and participant characteristics (e.g. age, setting, etc.). As data allowed, we compared quantitative results on

acceptability and preferences for VRs and other contraceptive, HIV prevention, and menopausal management technologies.

# Results

## Characteristics of included studies

We screened 1,110 unique citations in the published and grey literature (Fig 1). We excluded 772 citations yielding 338 citations for full text review. Of these, 68 reports from 47 unique studies met eligibility criteria. Most references that were excluded during full text review were conducted in high income countries or did not report acceptability outcomes. Most identified grey literature, including conference abstracts and reports, were duplicates of published literature and excluded.

Twenty-four observational studies and 14 RCTs were included from the peer-reviewed literature; nine additional observational studies and one RCT (the QUATRO study) were identified in the grey literature (Table 2). Eligible studies were conducted in 25 LMICs, with most studies from Asia (n = 17) and sub-Saharan Africa (n = 17). The most common VR indications were contraception (n = 28) and HIV prevention (n = 10). Nearly all studies were conducted among women of reproductive age (with varying definitions across studies; most definitions were 18 to 35 or 18 to 49 years) (n = 34), and postpartum/lactating women of reproductive age (n = 4). Most studies evaluated VR acceptability concurrent to use (n = 20), but some (n = 7) explored acceptability both prospectively (i.e. prior to use) and concurrent to use. Most RCTs had low risk of bias scores (n = 10), but observational studies largely had high or unclear risk of bias (n = 14).

## Acceptability

**Affective attitude–overall acceptability.** Overall assessments of affective attitude were reported in seven RCTs and nine observational studies that involved actual use of VRs (Table 3). In RCTs, acceptability ranged from 70.8 to 96.7% [32, 44, 58, 64]. One RCT

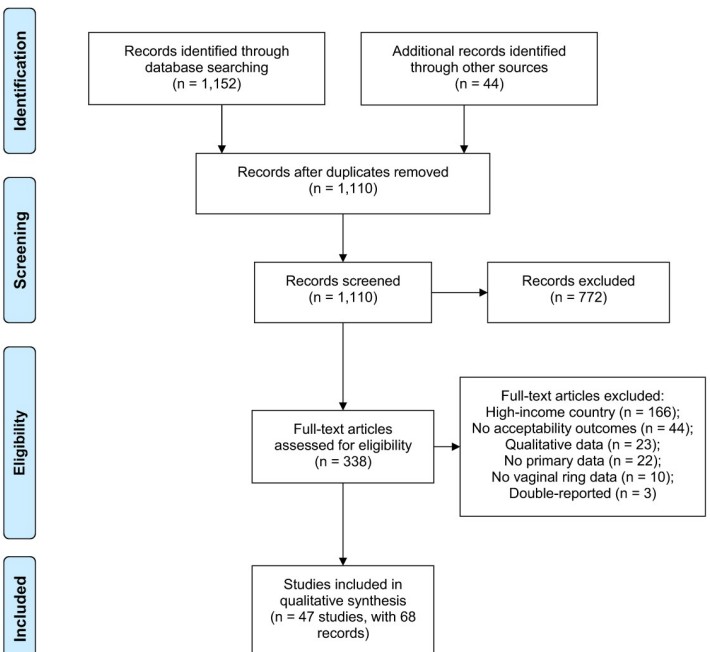

**Fig 1. PRISMA flow diagram.**

**Table 2. Study characteristics.**

| Characteristic | Randomized Controlled Trials N (%) | Observational Studies N (%) | Grey Literature[a] N (%) | Total Studies N (%) | References |
|---|---|---|---|---|---|
| **Total** | 14 (100%) | 24 (100%) | 9 (100%) | 47 (100%) | |
| **Location** | | | | | |
| Asia: China, India, Malaysia, Pakistan, Thailand | 5 (36%) | 10 (42%) | 2 (22%) | 17 (36%) | [26–43] |
| Sub-Saharan Africa: Kenya, Malawi, Nigeria, Rwanda, Senegal, South Africa, Tanzania, Uganda, Zambia, Zimbabwe | 7 (50%) | 5 (21%) | 5 (56%) | 17 (36%) | [9, 44–75] |
| Americas: Brazil, Chile[b], Dominican Republic, Mexico | 2 (14%) | 4 (17%) | 1 (11%) | 7 (15%) | [76–84] |
| Other: Egypt, Tunisia, Russia, Ukraine | - | 3 (13%) | 1 (11%) | 4 (9%) | [85–89] |
| Cross-regional[c] | - | 2 (8%) | - | 2 (4%) | [90–92] |
| **Indication** | | | | | |
| Contraception | 4 (29%) | 20 (83%) | 4 (44%) | 28 (60%) | [26, 27, 29–31, 33–36, 38–43, 50–52, 55, 59, 69, 70, 76–78, 80, 82, 84–92] |
| HIV-prevention | 4 (29%) | 2 (8%) | 4 (44%) | 10 (21%) | [9, 45–47, 54, 56, 57, 60–68, 71, 73, 74, 79] |
| Multipurpose prevention technology | 2 (14%) | 1 (4%) | 1 (11%) | 4 (9%) | [48, 49, 53, 58, 72, 75, 83] |
| Abnormal uterine bleeding | 3 (21%) | - | - | 3 (6%) | [28, 32, 44] |
| Chronic pelvic pain | 1 (7%) | - | - | 1 (2%) | [37] |
| Menopause symptom management | - | 1 (4%) | - | 1 (2%) | [81] |
| **Study population** | | | | | |
| Women of reproductive age | 14 (100%) | 16 (67%) | 4 (44%) | 34 (72%) | [9, 26, 28–34, 36, 37, 39, 40, 42, 44–47, 51–53, 55–61, 64–68, 72–75, 77, 79, 80, 82–88, 90–92] |
| Young women (18 to 21) | - | - | 2 (22%) | 2 (4%) | [67, 68, 71] |
| Postmenopausal women | - | 1 (4%) | 1 (11%) | 2 (4%) | [81] |
| Postpartum/lactating women of reproductive age | - | 3 (13%) | 1 (11%) | 4 (9%) | [27, 35, 50, 69, 70, 89] |
| Not reported | - | 4 (17%) | 2 (22%) | 6 (13%) | [38, 41, 43, 48, 49, 54, 62, 63, 76, 78] |
| **Timing of acceptability assessment** | | | | | |
| Prospective (hypothetical or prior to use) | - | 5 (21%) | 5 (56%) | 10 (21%) | [40, 48, 49, 53, 54, 57, 67, 68, 71, 82, 85, 86, 88, 91] |
| Concurrent to use | 7 (50%) | 11 (46%) | 2 (22%) | 20 (43%) | [26, 27, 29, 31, 32, 34–39, 42, 55, 59, 62–64, 77, 79, 80, 83, 92] |
| Prospective and concurrent | 4 (29%) | 1 (4%) | 2 (22%) | 7 (15%) | [9, 41, 43–47, 50, 56, 58, 60, 61, 65, 66, 69, 70, 72–75] |
| Prospective, concurrent, and retrospective | 1 (7%) | - | - | 1 (2%) | [51, 52] |
| Concurrent and retrospective | - | 1 (4%) | - | 1 (2%) | [81] |
| Studies reporting continuation/use only | 2 (14%) | 6 (25%) | - | 8 (17%) | [28, 30, 33, 76, 78, 84, 87, 89, 90] |
| **Risk of Bias** | | | | | |
| Low | 9 (64%) | 10 (42%) | 1 (11%) | 20 (43%) | [9, 27, 28, 30, 32, 36, 44–47, 51, 52, 55–58, 60, 61, 65, 66, 72–76, 78, 81, 83–88, 90] |
| High | 2 (14%) | 8 (33%) | - | 10 (21%) | [26, 29, 31, 33, 34, 42, 50, 59, 69, 70, 79, 92] |
| Unclear | 3 (21%) | 6 (25%) | 8 (89%) | 17 (36%) | [35, 37–41, 43, 48, 49, 53, 54, 62–64, 67, 68, 71, 77, 80, 82, 89, 91] |

[a] Represents studies exclusively reported in the grey literature; some grey literature, i.e. unpublished manuscripts, were associated with studies already documented in the peer-reviewed literature.

[b] Chile was classified as an LMIC at the time the study was conducted.

[c] Cross-regional studies also included sites in the above-listed countries, Colombia, and Cuba.

**Table 3. Studies reporting vaginal ring overall acceptability and/or continuation outcomes, endline assessment.**

| First author, year (Trial name) | n (ring users) (Country) | Indication | Ring attributes | Satisfy (%) | Recommend (%) | Acceptable (%) | Like (%) | Continuation by month/cycle (%) | Risk of bias[a] |
|---|---|---|---|---|---|---|---|---|---|
| **Randomized Controlled Trials** | | | | | | | | | |
| **Hardy; 2007 [79]** | 405 (Brazil) | HIV prevention | EE, flexible, transparent, colorless, 54x4 mm (NuvaRing) | - | - | - | 52.9 | - | H |
| **Nel; 2016 [64]** | 280 (Kenya, Malawi, South Africa, Tanzania) | HIV prevention | Dapivirine, platinum-catalyzed silicone, 56x7.7mm | - | - | 96 | - | - | U |
| **Kestelyn, Kestelyn; 2018, 2018 [51, 52]** | 120 (Rwanda) | Contraception | EE, flexible, transparent, colorless, 54x4 mm (NuvaRing) | - | 98.3[b] | | - | - | L |
| **Mohamed; 2011 [59]** | 300 (Egypt) | Contraception | EE, flexible, transparent, colorless, 54x4 mm (NuvaRing) | - | - | - | - | 3 mo: 87.7 6 mo: 82.7 9 mo: 81.3 12 mo: 79.9 | H |
| **Sharma; 2018 [39]** | 225 (India) | Contraception | EE, flexible, transparent, colorless, 54x4 mm (NuvaRing) | 95.3 | - | - | - | - | U |
| **Hashim; 2012 [44]** | 48 (Egypt) | Abnormal uterine bleeding | EE, flexible, transparent, colorless, 54x4 mm (NuvaRing) | 70.8 | - | - | - | 3 cycle: 100 | L |
| **Jain; 2016 [32]** | 30 (India) | Abnormal uterine bleeding | EE, flexible, transparent, colorless, 54x4 mm (NuvaRing) | 96.7 | 90 | - | - | 90 | L |
| **Minnis, Weinrib, van der Straten; 2018, 2018, 2018 [58, 72, 75] (TRIO)** | 277 (South Africa, Kenya) | MPT | Placebo, silicone elastomer (dimensions NR) | - | - | - | 65.6[c] | - | L |
| **Thurman; 2018 [83] (CONRAD A13-128)** | 27 (Dominican Republic) | MPT | G1: TDF G2: TDF+LNG G3: Placebo Hydrophilic polyurethane, 55x5.5mm | - | - | - | - | G1: 90.9% G2: 100% G3: 100% | l |
| **Observational Studies** | | | | | | | | | |
| **Barreiros, Guazzelli; 2007, 2009 [76, 78]** | 75 (Brazil) | Contraception | NR | - | - | - | - | 82.7 | L |
| **Buckshee; 1990 [26]** | 96 (India) | Contraceptive | LNG, Silastic, 55.6x9.5 mm | - | - | - | - | 52 wks: 44.5 | H |
| **Chen; 1998 [27]** | 197 (China) | Contraception | EE, flexible, transparent, colorless, 54x4 mm (NuvaRing) | - | - | - | - | 34.6 | U |
| **Das; 2016 [29]** | 50 (India) | Contraception | EE, flexible, transparent, colorless, 54x4 mm (NuvaRing) | 95 | 96 | 92 | - | 88 | H |
| **Faundes, Hardy; 1981, 1983 [77, 80]** | 355 (Brazil and Dominican Republic) | Contraception | LNG + estradiol, Silicone elastomer, 58 mm | - | 62.1 | - | - | - | U |
| **Gupta; 1986 [31]** | 70 (India) | Contraception | Progesterone, Silicone elastomer, 55.6x9.5 mm | - | - | - | - | 3 mo: 55.7 6 mo: 45.7 9 mo: 37.1 (calculated) | H |
| **Koetswang; 1990 [90]** | 789 in LMIC (Tunisia, Zambia, Russia, India, Thailand, Pakistan, Brazil, Colombia, Cuba) | Contraception | LNG, Silicone elastomer, 55.6x9.5 mm | - | - | - | - | Africa: 31.9 Asia: 40.1 China: 68.3 Latin America: 42.9 | L |

*(Continued)*

**Table 3.** (Continued)

| First author, year (Trial name) | n (ring users) (Country) | Indication | Ring attributes | Satisfy (%) | Recommend (%) | Acceptable (%) | Like (%) | Continuation by month/cycle (%) | Risk of bias[a] |
|---|---|---|---|---|---|---|---|---|---|
| **Mehta; 1981 [34]** | 39 (India) | Contraception | EE + d-norgestrel, polysiloxane, 61x9.5 mm | - | - | - | - | 56.4 | H |
| **Pandit; 2014 [36]** | 252 (India) | Contraception | EE, flexible, transparent, colorless, 54x4 mm (NuvaRing) | 94.2 | 93.2 | - | - | - | L |
| **RamaRao, RamaRao, Ishaku; 2015, 2015, 2018 [50, 69, 70]** | 363 (Kenya, Nigeria, Senegal) | Contraception | Silicone elastomer, 58mm x 8.4mm, 10 mg progesterone daily, administered continuously up to 3 months | 96.8 | 97.9 | - | - | - | H |
| **Santibenchakul; 2016 [38]** | 39 (Thailand) | Contraception | EE, flexible, transparent, colorless, 54x4 mm (NuvaRing) | 71 | 100 | - | - | 6 cycle: 97.4 | U |
| **Shaaban; 1991 [89]** | 103 (Egypt) | Contraception | Progesterone, silicone elastomer, 58.4x8.8 mm | - | - | - | - | 66.59 | U |
| **Sivin; 1981 [92]** | 1,636 (Brazil, Dominican Republic, Nigeria) | Contraception | Progesterone, silicone elastomer, 58x8.4 mm | - | - | - | - | Salvador, Brazil: 48 Campinas, Brazil: 55 Dominican Republic: 36 Nigeria: 38 | H |
| **Soni; 2013 [42]** | 184 (India) | Contraception | EE + ENG, ethinyl vinyl acetate, 54x4 mm | - | 97 | - | - | 3 mo: 94.6 12 mo: 86.4 | H |

EE = ethinyl estradiol; TDF = Tenofovir; LNG = levonorgestrel; ENG = etonogestrel; mo = months; G = Group; MPT = multipurpose prevention technology; wks = weeks

[a] Risk of bias summary assessments. H = high; L = low; U = unclear

[b] At end of study

[c] Mean score of liking product (1–5 Likert scale, higher score indicates higher acceptability)– 3.28, converted to 100 point scale.

comparing intermittent and continuous users of NuvaRing in Rwanda reported that at baseline <1% of women would recommend the VR; while, at the end of the study, 98% would recommend the VR [52]. Across studies, a smaller proportion of women reported 'liking' the VR (52.9 to 62.6%) [58, 79], with a higher proportion of users 'liking' the VR with increased duration of use [58, 75]. In observational studies, 'satisfaction' was mostly high, with most studies reporting satisfaction greater than 90% [29, 36, 70]. In one observational study, satisfaction was higher among women who completed at least two cycles of use compared to those who discontinued [70]. Five contraceptive observational studies in the past decade reported that 92 to 100% of users would recommend the VR or found it acceptable [29, 36, 38, 42, 70], whereas a smaller proportion of women recommended VRs in studies from the 1980s [77, 80]. VR acceptability findings were supported by conference proceedings [35, 43].

**Affective attitude–other.** Other affective attitude outcomes, such as 'willingness to use' and 'the best method' were generally only reported by one study. In the IPM 011 study for HIV prevention, 69% of women said they were 'very keen' to use the VR after first hearing about it; at the final visit, all women said they would be willing to use the VR if it were found to be effective [74]. In two recent RCTs for HIV prevention, 68 to 96% of users would consider or were likely to use the VR in the future [45, 60]; and, a descriptive study among young women in South Africa reported that 44% would hypothetically try the VR as an HIV

prevention product [71]. In the placebo TRIO study, most users reported it was acceptable to leave the VR in for one month (71.7%) and during menses (53.9%) [58]. In an RCT for chronic pelvic pain, 80% of VR users reported a composite outcome including compliance, acceptability, and that users would recommend the VR to others [37]. In an observational study of the VR for symptoms of menopause, 100% of women reported a positive experience [81]; and, an observational contraceptive study from the 1980s reported VR users were happy to have selected the VR (70%) and had a positive experience (64%) [77, 80].

**Affective attitude–physical attributes.** An important component of acceptability relates to physical attributes of the device. Several studies reported how women felt about ring size, appearance, color, and texture. In the TRIO study, two-thirds of women reported the VR looked acceptable [58]. In one RCT and two observational studies, 61.4 to 85.7% of users reported acceptable VR size [58, 70, 81]. In an observational study acceptability of VR color (88.3% baseline, 94.7% follow-up) and texture (53.2% baseline, 86.2% follow-up) increased with the duration of VR use [70]. Among women who did not find the VR acceptable, the proportion stating it was too big (45.7% baseline, 18.1% follow-up) or too soft (36.2% baseline, 9.6% follow-up) decreased with duration of use [70].

**Burden–ease of use.** Studies reporting data on burden reported ease of VR insertion and removal. In RCTs, greater than 85% of women reported ease of insertion across all time periods [52, 64]; others reported increasing ease of insertion or removal over time [36, 70, 74]. In observational studies, greater than 90% of women reported ease of insertion [29, 42, 70, 77, 80]; and, 89.3 to 98.5% reported ease of removal [42, 58, 70, 77, 80]. In observational studies, 12 to 14.1% of users reported difficult insertion/removal [38, 79].

**Burden–cognitive/emotional burden and social harms.** Studies measured the cognitive and/or emotional burden of VR use; however, there was no common measure reported across studies. Four RCTs of VR for HIV prevention and one NuvaRing RCT reported that at baseline or early in the study 8 to 43% of VR users had worries or concerns regarding partners not liking the ring or feeling the ring during sex, expulsions, loss in the body, discomfort during sex, or general worries, with concerns decreasing over time in nearly all studies [51, 64, 66, 73, 74] and varying between countries [47, 74]. In one RCT, worries were significantly higher among women who had experienced a partner-related social harm (defined as nonmedical adverse consequences from VR use or trial participation more generally) [66]. One observational study of contraceptive VR use in Brazil and the Dominican Republic reported that 34% of women worried whether the VR was inserted correctly [77, 80].

**Ethicality/Fit within value system.** While no studies directly measured ethicality, operationalized as how the VR fit within an individual's value system, several studies examined VR disclosure to partners and families, and partner/family support of VR use. Three RCTs for HIV prevention and NuvaRing reported 64.2 to 99.2% of study participants disclosed VR use to their partner [52, 74, 93]. IPM 011 reported that 59% of study participants would use the VR without telling their partner [74]; and, 62.6% of women in TRIO reported that it was possible to use the VR without partner knowledge [58]. The MTN-020/ASPIRE RCT reported that 12% of partners did not approve of ring use at the end of the study period [45]. An observational study reported that family and others were not likely to know that women used VR (85 to 92.5% across sites in Brazil and the Dominican Republic) [77, 80]; and, a discrete choice experiment reported that the VR could easily be used without family knowledge (80.3%) [58]. However, in an observational study, approximately 90% of users said that their family would support VR use [70]. And, the least common specific VR worry at baseline was it not being liked by their partner (4%) or family (10%) [45]. An unpublished observational study of dapivirine VR reported that 57% of women disclosed their trial participation to their partner, but 34% reported ring removals due to their partner's influence [63]. Grey literature with country-

specific results from Nigeria reported participants who said that their partner or family would support VR use were more satisfied with the VR (95%) compared to those whose partners or family would not support VR use (20%; p<0.01) [50, 70].

**Opportunity costs–menstrual bleeding.** Menstrual bleeding was the most frequently reported VR opportunity cost, with 11 studies (nine of which involved hormonal APIs) reporting on breakthrough bleeding, withdrawal bleeding, duration of menses, and other bleeding outcomes. RCTs largely reported improved bleeding outcomes (breakthrough bleeding, spotting, intended bleeding patterns) [28, 30, 32, 37, 59, 94]; comparable bleeding outcomes between active VRs and comparison products (Pictorial Blood Loss Assessment Chart scores, duration of menses, and intermenstrual and breakthrough bleeding) [28, 32, 64, 94]; or, no instances of abnormal vaginal bleeding [60]. An observational study reported that VR users had less frequent 'normal bleeding,' and significantly higher levels of amenorrhea compared to IUD users; however, it is unknown whether amenorrhea was acceptable to users [27]. In two observational studies from the 1980s and 1990s of contraceptive VRs, users reported more bleeding problems compared to combined oral contraceptive (COC) users [77, 80]; and, that resumption of menstruation was delayed in VR users compared to IUD users [89].

**Opportunity costs–sexual intercourse.** Thirteen studies reported outcomes related to sexual opportunity costs, most commonly reporting on the user or partner feeling the VR during sex. RCTs reported that 82.5 to 92% of users never felt the ring during sex [52, 64]. In three RCTs, many women (47.5 to 74%) reported that their partners did not feel the ring during sex; some (20 to 33%) reported that partners felt the ring during sex, but it wasn't a problem; a smaller proportion of women (19%) reported that their partners liked the feeling of the ring during sex; 13 to 16% of women didn't know if their partners felt the VR during sex; and, very few women reported that their partners (0 to 3%) did not like the way the VR felt during sex at least once during the study period [52, 64]. In one RCT for HIV prevention, 82% of women did not mind wearing a VR during sex and the VR was acceptable to 88% of partners [45]. Similarly, observational studies reported that the VR did not cause discomfort during sex [34] and was not felt during intercourse by 70 to 80% of women [29, 36, 70], although this sometimes changed over time [36, 70]. In the TRIO study, the ring felt acceptable to a majority of users (55.9%) and their partners (59.7%) during sex [58]. A range (3 to 40%) of partners or clients felt the VR during sex [29, 34, 38, 70]. In observational studies, the VR did not cause discomfort to partners [31]; most partners did not object to ring use [36]; and, 94% of partners did not object to the ring during sex [42]. There were no changes in sexual desire, sexual pleasure, or sexual activity in observational studies reporting these outcomes [27, 38, 59, 70].

**Opportunity costs–discharge.** Many studies reported non-API side effects of VR use related to vaginal discharge or vaginitis. All RCTs reported low levels of vaginal discharge and there were no significant differences between VR and comparison groups [28, 30, 32, 37, 52, 64, 74]. In seven additional observational studies for contraception and pelvic organ prolapse, 10 to 33% of women reported increased or excessive vaginal discharge [31, 38, 77, 80], and 2.5 to 3.8% of women reported vaginitis [29, 36, 42].

**Opportunity costs–vaginal comfort or discomfort/irritation.** Studies also reported outcomes related to comfort and discomfort/irritation. Two RCTs for HIV prevention reported VR 'comfort,' with 87 to 97% of women reporting that the VR is 'usually' or 'very' comfortable [45, 64, 74]. Few VR users (4.2 to 14%) reported vaginal discomfort or irritation across indications [26, 31, 36, 37, 94]. Feeling the VR varied across timepoints (0.6 to 21%) during daily activities in RCTs of HIV prevention [74] and contraception [44].

**Opportunity costs–expulsions and slippage.** Many studies reported outcomes related to VR expulsions and slippage. RCTs across indications reported expulsions in 0 to 12% of users [30, 32, 37, 44, 52, 95], primarily attributable to sex, urination or defecation [52, 61, 64, 96].

Observational studies also reported low proportions of women experiencing an expulsion across indications ranging from 1 to 10% [70, 81]. In one study, a larger proportion (33 to 50%) of women reported VR slippage at two time points [70], with reports of slipping and expulsions higher among women who discontinued (33 to 75% and 17 to 26%, respectively). In two studies of contraceptive VR from the 1980s in India, up to one-quarter of women experienced expulsions [31, 33].

**Perceived effectiveness.** No studies assessed perceived effectiveness of the VR platform separately from API effectiveness.

**Self-efficacy.** No peer-reviewed studies reported a measure of self-efficacy as a component of acceptability. In a currently unpublished study of the dapivirine VR, 31% of participants felt they could optimally support themselves to change the VR on time after 1 month of use [62].

## Continuation

Most trials and observational studies reporting continuation outcomes suggest the VR is largely acceptable to women across indications (Table 3) and more recent studies report higher rates of continuation compared to earlier studies. In contraceptive RCTs conducted after 2000, continuation rates ranged from 80 to 100% [30, 32, 44, 59], and in HIV prevention RCTS conducted in the same time period, continuation rates ranged from 85 to 100% [9, 64, 83]. Recent observational studies of contraceptive VR reported higher continuation rates (77 to 97.4%, higher during earlier months of use) [36, 38, 42] than older observational studies (31 to 66%) [26, 27, 34, 90]. Several observational studies reported that 77 to 93% women continued or wanted to continue using the VR after study completion [36, 43, 94]; in one study, this proportion was significantly greater than combined oral contraceptive users (25.5%) (P = 0.001) [44].

Women reported discontinuation due to both medical and personal reasons. Medical reasons for discontinuation included bleeding irregularities/menstrual problems for the contraceptive VR [26, 27, 29, 32, 34], expulsions [26, 27, 29, 34, 89], vaginitis/vaginal problems [26, 27, 38], adverse effects or use problems (not specified) [36, 77, 80, 89]; stress urinary incontinence [29], ring removal >48 hours [27], pelvic inflammatory disease [34], edema and tenderness of legs [34], changes in emotional state [34], ovarian cysts [26], involuntary pregnancy [26], coital problems [29], and other medical reasons not specified [27]. Personal reasons for discontinuation included planning pregnancy [26, 36, 42, 59, 89]; disliking the method [26, 27], perceived ineffectiveness [77, 80], foreign body sensation or discomfort [29], relocation [26, 42]; switching methods [42]; desire for surgical or other long lasting contraceptive method treatment [36]; general ring worries [73]; expense [36]; and, other personal reasons not specified [27, 34, 89].

## Values and preferences

Across studies, preference differed depending on available methods, geography, age, and product use experience. Between 1.8 to 52.9% of women chose or preferred the VR over other HIV prevention, contraception, and MPT products, with variation across countries [40, 57, 75, 77, 79, 80, 88, 97]. The TRIO study also reported variation in VR preference by age, with women 25 to 30 years three times more likely to prefer the VR compared to women 17 to 24 years (95%CI: 1.2, 8.2) [75]. Provider counseling also increased VR choice from pre- to post-counseling [40, 85, 88]. Studies that evaluated women's interest in hypothetical products indicated lower preferences for VRs than for other HIV prevention and MPT products, with significant variation by country and age, with adult women having higher preference for VRs [48, 49, 67, 68, 71].

In an African discrete choice experiment, 92% of women preferred dual prevention products for HIV and pregnancy versus products that only prevent HIV or pregnancy, with 20% of

participants' product choice dominated by HIV prevention efficacy and 44% by pregnancy prevention (59% in South Africa, 30% in Kenya, p<0.001) [57]. In a Quatro sub-study, efficacy was the most important product feature in HIV prevention (67%), followed by cost (14%), method of use (<10%), and location product is collected [46]. In the IPM 011 study, women reported they most liked that the VR might someday be used to prevent HIV and, 100% of product naïve women were willing to use the VR, if it was effective. [74].

In one study, most women (80%) preferred partner approval of the VR [74]; in another, women were less likely to try the VR if they thought their partner might not like the method [71]. Between 35 and 59% of women reported that it was important to be able to use a product without partner knowledge [57, 74]; and, 49 to 67% of women said it was important that their partner not feel the VR during sex [51, 64, 74]. However, in a QUATRO sub-study, partner awareness of method during sex was not a significant factor in product choice [46]. In the IMP 011 trial, women users liked that the VR did not interfere with 'normal, natural' sex, and least liked that the VR might come out or change the feeling during sex for the male partner [74]. An observational study of the progesterone VR reported that, among women who did not choose the VR for contraception, 17% thought it would be uncomfortable during sex [70].

In the TRIO study, women who chose to use the VR after trying three products reported that frequency of use (13.3%) and previous experience with vaginal insertion (46.7%) were attributes impacting their selection [75]. In another study, women who chose to use the weekly patch or daily COCs reported that they did not choose to use the VR because they were not comfortable inserting the VR (41 to 49%), and the VR was not easy to use (38 to 39%) [98]. In a phase I/II study of the dapivirine VR versus placebo VR for HIV-prevention, 96% of women preferred continuous daily (vs non-daily) VR use [64]. Similarly, in two multisite observational studies of contraceptive choice, three-quarters of women who choose the VR reported convenience and 66 to 72.4% reported monthly use as preferences [40, 98]. Among postpartum women in Africa, women who did not choose a progesterone VR explained that alternative methods were easier to use (26%), that they were uncomfortable with VR insertion (23%), or that the VR required repeat follow-up visits (15%) [70]. In an early observational study in Brazil and the Dominican Republic, 55.6% of users said that ease of use was the most liked characteristic of a contraceptive VR [77, 80]. Reports from the grey literature that were conducted with product naïve end-users about hypothetical HIV prevention and MPT products indicated that the frequency of use and the ability to remove the VR were reported to be major drivers of preference [49] and willingness to use [71].

In the TRIO study, 30% of VR users reported that safety/absence of side effects were an important product attribute [75]. A study of the progesterone VR for postpartum women reported that, among women who did not choose the VR, 20% were worried it could affect their health, but fewer than 5% worried the VR could affect the baby's health [70]. With hypothetical HIV prevention products among young women in South Africa, VRs with good safety and few side effects contributed to improved opinions about and willingness to try the method [71]. A pre- and post-counseling study reported that product-naïve women perceived that the VR did not have many side effects compared to other contraceptive methods [40].

In a cross-sectional study of contraceptive choice in Russia and Ukraine, 69 to 72% of VR choosers reported physician recommendation was an important choice factor; and 31 to 35% of non-VR choosers said it was because they did not know anybody who uses it (31 to 35%) [98]. Similarly, women chose an alternative method to the VR in Kenya, Nigeria and Senegal because they or someone else had used it before (37%) or they already knew about the method (30%) [70]. Grey literature similarly reported preferences for the VR increase with familiarity, and that non-users were interested in the VR, but experienced misperceptions or concerns.

In the TRIO study, 20% of VR users said the most important product attribute was availability/access [75]. A study of the progesterone VR for postpartum contraception in Kenya, Nigeria, and Senegal reported that, among women who did not choose the VR, 15% did so because the VR required repeat follow-up visits [70].

## Discussion

This systematic review investigated VR acceptability and related preferences among women in LMIC to inform further investment and/or guidance on VRs. Overall, VRs were very acceptable to women users across indications, with high rates of satisfaction and willingness to recommend VRs to others. Most women found VRs easy to insert/remove; and, although some women had concerns, these typically decreased, and most women reported acceptability of side effects and product features. Few studies explored differences by participant age or region, limiting our ability to draw conclusions regarding variations in acceptability by these factors.

The current review found that initial and hypothetical opinions regarding the VR were lower, but that acceptability increased as women gained use experience over several months. This experience-effect has also been reported in RCTs of VR acceptability in high-income countries [99]. While VRs are long-acting (1 month to 1 year duration), women need to be comfortable with vaginal insertion and removal; in several studies, self- reported lack of comfort with insertion was a factor associated with not choosing the VR [70, 88]. Previously, vaginal practices such as tampon use have been found to be associated with increased acceptability of vaginal products, possibly due to increased familiarity with touching the vagina and/or vaginal insertion [100]. High-quality training, counseling, and education may enhance uptake and continuation in the face of lack of comfort with vaginal insertion, VR unfamiliarity, and/or initial concerns with the VR [101]. In several studies, higher proportions of women reported preferring the VR after watching informational videos or receiving provider counseling [60, 82]. Peer support has also been noted as influential in helping women air concerns about ring use, overcome fears and learn strategies about ring use and disclosure [102–104].

Acceptability, study continuation, and continued VR use were higher in studies occurring in the 2000s compared to studies from the 1980s and 1990s. This trend may demonstrate growing social acceptance of the VR or knowing others using the VR [40, 70]; physicians' recommendations of VRs [88]; and/or, changes in norms regarding the use of vaginal products [105]. None of the studies in this review directly assessed these broader social and community-level factors, although one noted that the introduction of VRs without education or information campaigns to promote use may have influenced user acceptability [77, 80]. Changes in VR physical attributes may also contribute to temporal differences in acceptability, with early studies typically using larger rings made from different materials compared to more recent studies of the smaller NuvaRing or use of more flexible rings.

Burden and opportunity costs of using VRs were largely acceptable to women. For example, VRs had acceptable bleeding patterns that were considered consistent with or preferable to those of other hormonal contraceptives, possibly due to lower systemic exposure to hormones reducing side effects [1, 2, 106]. Use of VRs during sexual intercourse was generally acceptable; women liked that VRs did not interfere with sex and had minimal impact on libido or sexual activity, although interference with sex was a salient initial concern with this platform, and some ring users removed the ring to avoid ring discovery or partner-related challenges [73, 107]. While women reported concerns and experiences with VR expulsions and slippage, some VRs have been successfully formulated to minimize expulsions and slippage. Issues with foreign body sensation, and/or discomfort, varied widely across studies. Cognitive burden may be a particularly important consideration for new VR users, with concerns decreasing as

women gain experience. The nature and severity of VR concerns vary by setting and user experience. Questions remain about differences in VR users across age groups and settings regarding each of the issues mentioned above, as well as the impact of misconceptions (e.g. fears of VR loss in the body).

As efforts to develop single and multipurpose VRs advance, achieving 'real-world' use necessitates the consideration of women's values and preferences [108]. The current review suggests a preference for methods with less frequent dosing [57, 75, 98], a finding supported by qualitative preference data from research on contraceptives [109], as well as data demonstrating a trend of increasing adherence as the dosing interval is decreased [110]. Women reported valuing lighter periods, regular menstruation, and the steady low hormone levels provided by VRs [34, 57, 98]. When asked, women also expressed a preference for MPTs that prevent both HIV and pregnancy, although the strength of this preference varied by country and product experience [57]. While partner approval of products was preferred, so was the ability to use the product without a partner's knowledge and non-interference with sex. Despite discretion, the VR may provide lower potential for discreet product use because it may be felt by a male partner during sex. It is critical to recognize that platform preferences may not be generalizable within or across age groups and settings, emphasizing the importance of a range of available drug-delivery platforms to align with individual women's values and preferences.

The current review was limited by heterogeneity between studies and a lack of standardization of acceptability outcome measures. We found that dimensions of affective attitude, burden and opportunity costs were frequently examined; however, ethicality, intervention coherence, perceived effectiveness, self-efficacy, and broader values and preferences remain relatively unexplored. We suggest development of generic, standardized acceptability measures mapped to the theoretical framework of acceptability constructs [21], using established methods to develop patient reported outcome instruments [111]. In the current review, we found several acceptability outcomes such as 'disclose of use to partner or family' and 'partner or family approval' that do not clearly map to the Sekhon acceptability framework, possibly indicating that an additional construct capturing 'social' or 'normative' acceptability could be important, particularly for potentially stigmatized conditions, such as sexual and reproductive health, mental health, or obesity. Finally, while qualitative acceptability data were not included in the current review, it is possible that more detailed data regarding acceptability constructs, not frequently reported in quantitative data, exist in qualitative studies of vaginal ring acceptability. For example, in a qualitative analysis of vaginal ring acceptability in a trial for HIV prevention, use of the ring appeared to give women a sense of ownership over HIV protection, which would be challenging to capture in a structured questionnaire [102].

While we wanted to investigate VR acceptability across indications, it was challenging to distinguish the influence of the API on reports of adverse events and side effects, discomfort, or satisfaction and acceptability when assessing studies using VRs with active products. We excluded adverse events, side effects, or negative effects known to be related to the API where possible; however, users' reports of satisfaction were likely influenced by their experience of API side effects, efficacy, and other dependent experiences. RCTs generally had a low risk of bias, with most having low reported attrition rates (median: 4.1; range 0–19.3); observational studies had varying risk of bias scores, with primary issues across studies being selection bias, non-consecutive inclusion of participants, and inadequate description of participants to replicate the study. There was no indication that findings differed by risk of bias assessment. Further, most studies did not use acceptability as their primary outcomes, possibly reducing the relevance of the bias assessment and underpowered acceptability outcomes, in some cases. Future acceptability research should 1) utilize theoretical constructs of product acceptability to assess multiple domains, which would allow greater insight into VR features and context

contributing to acceptability; 2) use standardized acceptability measures to improve comparability across studies and more meaningfully contribute to the evidence base; 3) disaggregate acceptability by age, parity, or other demographic variables to elucidate VR acceptability by key participant characteristics; and 4) examine and integrate qualitative vaginal ring acceptability data, as well as data from high income countries. We are currently conducting a related systematic review of vaginal ring acceptability in high income countries, which will further elucidate vaginal ring acceptability relevant to all women.

## Conclusions

The evidence strongly suggests that most women find VRs acceptable and easy to use, despite reported cognitive/emotional burden, impacts on sexual intercourse, and some issues with expulsions. An experience-effect was seen, with acceptability increasing over time as users gained experience with the method, and continuation was high across studies. Some women value features associated with the VR, including low dosing frequency, positive menstrual bleeding patterns, and the potential for multipurpose technologies. As such, the VR may play an important role in expanding sexual and reproductive health options for women.

## Supporting information

**S1 File. Search strategy for PubMed, Embase, and Web of Science.**
(DOCX)

**S2 File. Study characteristics, acceptability outcomes, and risk of bias assessment for studies in the peer-reviewed literature.**
(DOCX)

**S3 File. Study characteristics, acceptability outcomes, and risk of bias assessment for studies reported in conference abstracts and the grey literature.**
(DOCX)

## Acknowledgments

Co-authors JG, EM, and AS were supported by TIP, a program made possible by the generous support of the American people through the U.S. President's Emergency Plan for AIDS Relief (https://www.pepfar.gov/) to RTI International under the terms of the Grant No. AID-OAA-A-14-00012. Co-authors KR, KT, and JP were supported by the OPTIONS Consortium, a program made possible by the generous assistance from the American people through the U.S. Agency for International Development (USAID) and the U.S. President's Emergency Plan for AIDS Relief (PEPFAR). Financial assistance was provided by USAID (https://www.usaid.gov/) to FHI 360, the Wits Reproductive Health and HIV Institute, and AVAC under the terms of Cooperative Agreement No. AID-OAA-A-15-00035. The contents do not necessarily reflect the views of USAID, PEPFAR, or the United States Government. The funders did not play any role in the study design, data collection and analysis, decision to publish, or preparation of the manuscript.

## Author Contributions

**Conceptualization:** Jennifer B. Griffin, Kathleen Ridgeway, Elizabeth Montgomery, Kristine Torjesen, Rachel Baggaley, Ariane van der Straten.

**Data curation:** Kathleen Ridgeway, Rachel Clark, Jill Peterson.

**Formal analysis:** Jennifer B. Griffin.

**Funding acquisition:** Kristine Torjesen, Ariane van der Straten.

**Methodology:** Jennifer B. Griffin, Kathleen Ridgeway.

**Writing – original draft:** Jennifer B. Griffin, Kathleen Ridgeway.

**Writing – review & editing:** Jennifer B. Griffin, Kathleen Ridgeway, Elizabeth Montgomery, Kristine Torjesen, Rachel Clark, Jill Peterson, Rachel Baggaley, Ariane van der Straten.

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
