## [Decision Letter · Decision Letter 0]

7 Oct 2019

PONE-D-19-20130

Vaginal ring acceptability and related preferences among women in low- and middle-income countries: A systematic review and narrative synthesis

PLOS ONE

Dear Dr. Griffin,

Thank you for submitting your manuscript to PLOS ONE. After careful consideration, we feel that it has merit but does not fully meet PLOS ONE’s publication criteria as it currently stands. Therefore, we invite you to submit a revised version of the manuscript that addresses the points raised during the review process.

We would appreciate receiving your revised manuscript by Nov 21 2019 11:59PM. To enhance the reproducibility of your results, we recommend that if applicable you deposit your laboratory protocols in protocols.io, where a protocol can be assigned its own identifier (DOI) such that it can be cited independently in the future. For instructions see: http://journals.plos.org/plosone/s/submission-guidelines#loc-laboratory-protocols

We look forward to receiving your revised manuscript.

Kind regards,

Manuela De Allegri

Academic Editor

PLOS ONE

Journal Requirements:

2. In the Methods, please describe the procedure by which you assessed bias and heterogeneity.

'The authors have declared that no competing interests exist.'

We note that one or more of the authors are employed by a commercial company: RTI International.

Additional Editor Comments (if provided):

Reviewers' comments:

Reviewer's Responses to Questions

**Comments to the Author**

1. Is the manuscript technically sound, and do the data support the conclusions?

Reviewer #1: Yes

Reviewer #2: Yes

2. Has the statistical analysis been performed appropriately and rigorously? 

Reviewer #1: I Don't Know

Reviewer #2: Yes

3. Have the authors made all data underlying the findings in their manuscript fully available?

Reviewer #1: Yes

Reviewer #2: Yes

4. Is the manuscript presented in an intelligible fashion and written in standard English?

Reviewer #1: Yes

Reviewer #2: Yes

5. Review Comments to the Author

Reviewer #1: In this paper, the authors present a systematic review of literature related to the acceptability of vaginal rings, used for a variety of purposes, in LMICs. The paper is well written and an important contribution to the literature in the current context of emerging multipurpose vaginal ring technologies. I recommend that this paper is published.

A few things to consider:

How does including hypothetical acceptability, as well as studies that reported acceptability prospectively and concurrently impact the results? The authors state that acceptability increased over use but how was the overall acceptability score (included in Table 3) determined for studies when acceptability changed after/during use?

I also wonder if/how the wider context of contraceptive availability impacted the scores. For example, the data presented from Rwanda on lines 204-206, was the ring available in the context and/or a widely known method prior to use? Put differently, was the low baseline acceptability due to people not knowing about the ring because it was not common in the context or because they did not think they would like this type of contraceptive? I think that is an important distinction, although likely difficult to assess. This overarching point is briefly mentioned in lines 432-434 – preference increases with familiarity.

Non-API side effects are mentioned on line 130 but where are these reported? Are these considered under values and preferences (see line 138 and 419-426) or under opportunity costs (e.g. discharge and/or discomfort/irritation) – see lines 317-329?

Over half of the studies have high or unclear bias, how does this impact the acceptability findings?

The point in the discussion section about future studies using standardized measures is important. While it might be beyond the scope of this paper, any suggestions as to how to standardize some of the constructs that were not reported often (e.g. measures listed on 502-503)? I realize that qualitative studies were excluded from this review but I wonder if any of the qualitative studies that were excluded had more detailed information on some of these topics (e.g. ethicality).

Reviewer #2: This is a clear and valuable review. My main concern is your decision to exclude qualitative research. Considering your research interest in acceptability and values and preferences, and your use of narrative synthesis, it seems that including any qualitative research on this topic would have be valuable.

Other concerns are noted as follows:

- Your discussion of risk of bias is quite short and general – you only mention that you used “standardized methods”. What were these methods? It would be useful to understand what you considered bias to be and how exactly you assessed risk. In addition, how did you handle or account for studies with high or unclear risk of bias when considering their findings, considering you did not remove them?

- You focused on women in LMICs – could you justify this decision a bit more? You mention that unintended pregnancies are more common, as is the risk of HIV, so vaginal rings may be particularly useful in LMICs. But in terms of vaginal ring acceptability, why do you think this would be different between women in LMIC versus HIC contexts? Couldn’t we learn from women’s experiences in HICs as well? There is generally a lot more research from HICs on all topics so I worry that we missed including important insights that would be relevant to all women, simply because the study took place in a HIC. In addition, you note that there hasn’t been any review from HICs, so it would have made sense – in my opinion – to have included HIC literature too.

6. PLOS authors have the option to publish the peer review history of their article (what does this mean?). If published, this will include your full peer review and any attached files.

Reviewer #1: No

Reviewer #2: No

---

## [Author Response · Author response to Decision Letter 0]

18 Oct 2019

Comments to the Author

1. Is the manuscript technically sound, and do the data support the conclusions?

Reviewer #1: Yes

Reviewer #2: Yes

We thank the reviewers for these comments. 

2. Has the statistical analysis been performed appropriately and rigorously? 

Reviewer #1: I Don't Know

Reviewer #2: Yes

We thank the reviewers for these comments. 

3. Have the authors made all data underlying the findings in their manuscript fully available?

Reviewer #1: Yes

Reviewer #2: Yes

We thank the reviewers for these comments. 

4. Is the manuscript presented in an intelligible fashion and written in standard English?

Reviewer #1: Yes

Reviewer #2: Yes

We thank the reviewers for these comments. 

5. Review Comments to the Author

Reviewer #1: In this paper, the authors present a systematic review of literature related to the acceptability of vaginal rings, used for a variety of purposes, in LMICs. The paper is well written and an important contribution to the literature in the current context of emerging multipurpose vaginal ring technologies. I recommend that this paper is published.

A few things to consider:

How does including hypothetical acceptability, as well as studies that reported acceptability prospectively and concurrently impact the results? The authors state that acceptability increased over use but how was the overall acceptability score (included in Table 3) determined for studies when acceptability changed after/during use?

To avoid conflating hypothetical acceptability with findings related to actual use experience, we distinguished all hypothetical outcomes as such in the Results section of the manuscript. The studies that assessed hypothetical acceptability were primarily focused on potential users' values and preferences, and studies that did assess acceptability reported outcomes that were distinct from those reported by studies involving actual use of VRs. In Table 3, we report acceptability at the final assessment timepoint, and all studies included in this table assessed women's experiences with actual rings. We have clarified this in the results, as follows: 

Results Lines 231-232:

Overall assessments of affective attitude were reported in seven RCTs and nine observational studies that involved actual use of VRs. 

Results Lines 244-245:

Table 3. Studies reporting vaginal ring overall acceptability and/or continuation outcomes, endline assessment.

I also wonder if/how the wider context of contraceptive availability impacted the scores. For example, the data presented from Rwanda on lines 204-206, was the ring available in the context and/or a widely known method prior to use? Put differently, was the low baseline acceptability due to people not knowing about the ring because it was not common in the context or because they did not think they would like this type of contraceptive? I think that is an important distinction, although likely difficult to assess. This overarching point is briefly mentioned in lines 432-434 – preference increases with familiarity.

We appreciate this comment and agree that although few studies assessed broader community-level awareness or acceptance of VRs, general availability and awareness of VRs likely influenced users' perceptions and assessment of the VR. We have added the following text to the discussion: 

Discussion lines 513-515:

None of the studies in this review directly assessed these broader social and community-level factors, although one noted that the introduction of VRs without education or information campaigns to promote use may have influenced user acceptability. 

Discussion lines 558-563:

In the current review, we found several acceptability outcomes such as ‘disclose of use to partner or family’ and ‘partner or family approval’ that do not clearly map to the Sekhon acceptability framework, possibly indicating that an additional construct capturing ‘social’ or ‘normative’ acceptability could be important, particularly for potentially stigmatized conditions, such as sexual and reproductive health, mental health, or obesity.

Non-API side effects are mentioned on line 130 but where are these reported? Are these considered under values and preferences (see line 138 and 419-426) or under opportunity costs (e.g. discharge and/or discomfort/irritation) – see lines 317-329?

We have amended line 354 to more clearly indicate that vaginal discharge and vaginitis were non-API side effects, as follows: 

Many studies reported non-API side effects of VR use related to vaginal discharge or vaginitis. 

Over half of the studies have high or unclear bias, how does this impact the acceptability findings?

During the analysis, we evaluated whether and how the inclusion of studies with high or unclear risk of bias would influence findings of this review. We noted that most RCTs had low risk of bias but did not see any clear indication that findings differed by risk of bias assessment. Including all eligible studies allowed for broader representation of country settings, timeframes, and study designs, which we judged to be important to the objective of our review. We have added a line in the methods and discussion to capture this as follows:

Methods lines 183-184: 

No studies were excluded based on risk of bias; however, we did compare acceptability outcomes in low risk versus and high/unclear risk studies. 

Discussion line 576: 

There was no indication that findings differed by risk of bias assessment.

We also have included a more detailed description of the methods used in the assessment of risk of bias in lines 176-184, as follows:

We assessed risk of bias across randomized studies using Cochrane Collaboration methods by one primary and one secondary reviewer (KR and RC) against key criteria (22, 23). The following judgments were used: low risk, high risk, unclear (i.e. lack of information or uncertainty over bias potential). Risk of bias across non-randomized studies were similarly assessed (24). Conflicts were resolved via consensus. Studies with no ‘high risk of bias’ ratings were considered low risk of bias; studies with any ‘high risk of bias’ ratings were considered moderate risk of bias; and, two or more ‘high risk of bias’ ratings were considered high risk of bias. No studies were excluded based on risk of bias; however, we did compare acceptability outcomes in low risk versus and high/unclear risk studies.

The point in the discussion section about future studies using standardized measures is important. While it might be beyond the scope of this paper, any suggestions as to how to standardize some of the constructs that were not reported often (e.g. measures listed on 502-503)? I realize that qualitative studies were excluded from this review but I wonder if any of the qualitative studies that were excluded had more detailed information on some of these topics (e.g. ethicality).

We agree with the reviewer that further discussion of standardization of acceptability constructs is appropriate. We also believe that there may be more detailed information on some of these topics in qualitative studies of vaginal ring acceptability. 

In order to address these issues, we have added the following text to the discussion in lines 556-568, as follows:

We suggest development of generic, standardized acceptability measures mapped to the theoretical framework of acceptability constructs (21), using established methods to develop patient reported outcome instruments (112). In the current review, we found several acceptability outcomes such as ‘disclose of use to partner or family’ and ‘partner or family approval’ that do not clearly map to the Sekhon acceptability framework, possibly indicating that an additional construct capturing ‘social’ or ‘normative’ acceptability could be important, particularly for potentially stigmatized conditions, such as sexual and reproductive health, mental health, or obesity. Finally, while qualitative acceptability data were not included in the current review, it is possible that more detailed data regarding acceptability constructs, not frequently reported in quantitative data, exist in qualitative studies of vaginal ring acceptability. For example, in a qualitative analysis of vaginal ring acceptability in a trial of HIV prevention, use of the ring seemed to give women a sense of ownership over HIV protection, which would be challenging to capture in a structured questionnaire (103).

Reviewer #2: This is a clear and valuable review. My main concern is your decision to exclude qualitative research. Considering your research interest in acceptability and values and preferences, and your use of narrative synthesis, it seems that including any qualitative research on this topic would have be valuable.

We appreciate this comment from the reviewer and agree that the evaluation of qualitative data would be an important contribution to the body of work examining acceptability of vaginal rings. Our team decided to focus our efforts on quantitative data for two reasons:

a) Methods for conducting mixed quantitative/qualitative systematic reviews are complex and, to date, poorly defined compared to qualitative or quantitative reviews. (see e.g. Tricco AC, Antony J, Soobiah C, Kastner M, MacDonald H, Cogo E, Lillie E, Tran J, Straus SE. Knowledge synthesis methods for integrating qualitative and quantitative data: a scoping review reveals poor operationalization of the methodological steps. Journal of Clinical Epidemiology. 2016 May 1;73:29-35.)

b) Conducting a complex quantitative/qualitative review would have created additional financial and time costs. We wanted to ensure relevant vaginal ring acceptability data were published prior to the deliberations of the European Medicines Agency (EMA) on dapivirine vaginal rings (decision expected in Q2, 2020).

In the discussion of the manuscript, we have added a justification of our focus on quantitative data and encourage the future evaluation of qualitative vaginal ring acceptability data in methods and the discussion, as follows:

Methods lines 126-128:

We focused this review on quantitative studies due to the relatively complex and poorly defined methods around the integration of qualitative and quantitative data (22).

Discussion lines 564-567: 

Finally, while qualitative acceptability data were not included in the current review, it is possible that more detailed data regarding acceptability constructs, not frequently reported in quantitative data, exist in qualitative studies of vaginal ring acceptability.

Discussion lines 586-587: 

and 4) examine and integrate qualitative vaginal ring acceptability data, as well as data from high income countries.

Other concerns are noted as follows:

- Your discussion of risk of bias is quite short and general – you only mention that you used “standardized methods”. What were these methods? It would be useful to understand what you considered bias to be and how exactly you assessed risk. In addition, how did you handle or account for studies with high or unclear risk of bias when considering their findings, considering you did not remove them?

We thank the reviewer for this comment. We have included a more detailed description of the methods used in the assessment of risk of bias in lines 173-181, as follows:

We assessed risk of bias across randomized studies using Cochrane Collaboration methods by one primary and one secondary reviewer (KR and RC) against key criteria (22, 23). The following judgements were used: low risk, high risk, unclear (i.e. lack of information or uncertainty over bias potential). Risk of bias across non-randomized studies were similarly assessed (24). Conflicts were resolved via consensus. Studies with no ‘high risk of bias’ ratings were considered low risk of bias; studies with any ‘high risk of bias’ ratings were considered moderate risk of bias; and, two or more ‘high risk of bias’ ratings were considered high risk of bias. No studies were excluded based on risk of bias; however, we did compare acceptability outcomes in low risk versus and high/unclear risk studies.

Additionally, in our discussion of bias in the discussion (line 578), we added the following text:

There was no indication that findings differed by risk of bias assessment.

- You focused on women in LMICs – could you justify this decision a bit more? You mention that unintended pregnancies are more common, as is the risk of HIV, so vaginal rings may be particularly useful in LMICs. But in terms of vaginal ring acceptability, why do you think this would be different between women in LMIC versus HIC contexts? Couldn’t we learn from women’s experiences in HICs as well? There is generally a lot more research from HICs on all topics so I worry that we missed including important insights that would be relevant to all women, simply because the study took place in a HIC. In addition, you note that there hasn’t been any review from HICs, so it would have made sense – in my opinion – to have included HIC literature too.

We appreciate this comment from the reviewer and agree that the evaluation of vaginal ring acceptability in high income countries would be an important contribution. Out team decided to split the work on vaginal ring acceptability into two discrete reviews to maintain a manageable scope of work in the face of extensive and heterogeneous vaginal ring acceptability data; and, to facilitate publication of the most relevant acceptability data in advance of the European Medicines Agency (EMA) decision on dapivirine vaginal rings. We are currently conducting a second systematic review of vaginal ring acceptability in HICs, which is registered in PROSPERO (Review ID 150229, currently under assessment by editorial team).

In the discussion of the manuscript, we have added a justification of our focus on women in LMIC in lines 587-589, as follows:

We are currently conducting a related systematic review of vaginal ring acceptability in high income countries, which will further elucidate vaginal ring acceptability relevant to all women.

---

## [Editor Report · Decision Letter 1]

24 Oct 2019

Vaginal ring acceptability and related preferences among women in low- and middle-income countries: A systematic review and narrative synthesis

PONE-D-19-20130R1

Dear Dr. Griffin,

We are pleased to inform you that your manuscript has been judged scientifically suitable for publication and will be formally accepted for publication once it complies with all outstanding technical requirements.

With kind regards,

José das Neves

Academic Editor

PLOS ONE
---

## [Editor Report · Acceptance letter]

29 Oct 2019

PONE-D-19-20130R1 

Vaginal ring acceptability and related preferences among women in low- and middle-income countries: A systematic review and narrative synthesis 

Dear Dr. Griffin:

I am pleased to inform you that your manuscript has been deemed suitable for publication in PLOS ONE. Congratulations! Your manuscript is now with our production department. 

With kind regards,

on behalf of

Dr. José das Neves 

Academic Editor

PLOS ONE